# Breathing distortions in the metallic, antiferromagnetic phase of LaNiO$_3$

**Alaska Subedi$^{1,2}$**

**1** Centre de Physique Théorique, École Polytechnique, CNRS,
Université Paris-Saclay, F-91128 Palaiseau, France
**2** Collège de France, 11 place Marcelin Berthelot, 75005 Paris, France

## Abstract

I study the structural and magnetic instabilities in LaNiO$_3$ using density functional theory calculations. From the non-spin-polarized structural relaxations, I find that several structures with different Glazer tilts lie close in energy. The $Pnma$ structure is marginally favored compared to the $R\bar{3}c$ structure in my calculations, suggesting the presence of finite-temperature structural fluctuations and a possible proximity to a structural quantum critical point. In the spin-polarized relaxations, both structures exhibit the ↑0↓0 antiferromagnetic ordering with a rock-salt arrangement of the octahedral breathing distortions. The energy gain due to the breathing distortions is larger than that due to the antiferromagnetic ordering. These phases are semimetallic with small three-dimensional Fermi pockets, which is largely consistent with the recent observation of the coexistence of antiferromagnetism and metallicity in LaNiO$_3$ single crystals by Guo *et al.* [Nat. Commun. 9, 43 (2018)].


# 1 Introduction

The rare-earth nickelates $R\text{NiO}_3$ have received an enduring interest over the last two and half decades because they exhibit unique structural and electronic transitions that can be tuned from 600 to 130 K as a function of the rare-earth ion $R$ [1–3]. All rare-earth nickelates except LaNiO$_3$ [4] occur in a perovskite-type orthorhombic structure with the space group $Pnma$ in the high-temperature phase, which is metallic [5]. As the temperature is lowered, all orthorhombic rare-earth nickelates undergo a structural transition to a monoclinic structure with the space group $P2_1/n$ [6–9]. This structural transition involves a splitting of the Ni positions into two inequivalent sites, and it simultaneously transforms these materials into insulators. Additionally, these materials also undergo a magnetic transition that coincides with the structural and electronic transition temperature for $R = $ Pr and Nd, but occurs at a lower temperature for the rare-earth ions with smaller radii.

Although the electronic and magnetic transitions in the orthorhombic rare-earth nickelates show relatively large responses in the resistivity and susceptibility measurements, identifying the order parameters and the microscopic mechanism behind these transitions has remained challenging. The metal-insulator and paramagnetic-antiferromagnetic transitions in these materials were observed as early as 1989 [5,10]. However, the monoclinic $P2_1/n$ structure of the low-temperature phase was only resolved in 1999 [6,7]. The magnetic ordering occurs at the wave vector $(\frac{1}{2}, \frac{1}{2}, 0)_o$ relative to the orthorhombic unit cell [11–13]. But several arrangements of the magnetic moments are consistent with the available neutron scattering data, and the magnetic structure of the ordered phase has not been fully determined.

The metal-insulator transition in the orthorhombic rare-earth nickelates was initially believed to be a Mott transition [5]. However, the observation of the compression and expansion of the NiO$_6$ octahedra in an alternating manner in the low-temperature monoclinic phase makes this explanation untenable and, instead, points towards a charge ordering mechanism [6]. The Ni$^{3+}$ ions have a nominal occupancy of $e_g^1$ in these materials, and the absence of Jahn-Teller distortion for this electronic configuration is also surprising. Goodenough and Raccah have argued that this absence is due to a large covalency between the Ni 3$d$ and O 2$p$ orbitals, which makes the antibonding $e_g^1$ state highly delocalized [4,14]. Hartree-Fock cluster calculations by Mizokawa *et al.* that took into account the large Ni 3$d$–O 2$p$ covalency found breathing distortions of the O ions to be stable for the small rare-earth ion nickelates [15]. However, for larger rare-earth nickelates such as PrNiO$_3$ and NdNiO$_3$, they found the displacement of the Ni ions along the cubic diagonal direction to be favorable, which was not observed in high-resolution diffraction experiments [16,17].

The presence of a large Ni 3$d$–O 2$p$ covalency may explain the lack of Jahn-Teller distortion in the rare-earth nickelates, but this concept does not identify the microscopic instability that causes the breathing distortions in all the rare-earth nickelates except LaNiO$_3$. A major leap in understanding the phase transitions in the rare-earth niceklates was achieved by the insight of Mazin *et al.*, who showed that a charge ordering of the type $2e_g^1 \to e_g^0 + e_g^2$ occurs when the on-site Hund's rule coupling $J$ overcomes the on-site Coulomb repulsion (i.e., when $U - 3J$ is small) [18]. This charge ordering of the antibonding $e_g$ electrons naturally leads to an expansion and compression of the alternate NiO$_6$ octahedra that is experimentally observed in the low-temperature monoclinic phase of the rare-earth nickelates. Additionally, it also gives rise to the magnetic ordering ↑0↓0 (antiferromagnetic ordering of the Ni moments in the larger octahedra and absence of moments at the Ni sites in the smaller octahedra), which is consistent with the available neutron scattering data.

The mechanism for the phase transitions in the rare-earth nickelates suggested by Mazin *et al.*'s phenomenological model and density functional theory (DFT) calculations has been further supported by dynamical mean field theory (DMFT) calculations using realistic electronic

structures that span both the high-energy Ni $3d$ + O $2p$ [19] and low-energy antibonding $e_g$ energy scales [20]. Other theoretical studies using diverse techniques have also supported this description of the phase transition [21–24]. The DMFT calculations utilizing the low-energy $e_g$ manifold further highlighted the essential role played by the breathing distortions in causing the metal-insulator transition [20]. It was found that even a small breathing distortion splits the quarter filled $e_g$ bands, resulting in a system with a manifold of narrow half-filled bands. Alternatively, in the real space picture, the distortion makes the on-site energies of the neighboring quarter-filled $e_g$ states inequivalent, which causes one site to be half-filled and another to be empty. This change in the electronic structure, viewed from either picture, makes the system highly susceptible to undergo a transition to an insulating phase.

Moving the focus to the title compound $LaNiO_3$, it is curious that it shows a behavior that is distinct from all other rare-earth nickelates even though the ionic radius of $La^{3+}$ is close to that of the early members of the lanthanide series such as $Pr^{3+}$. $LaNiO_3$ occurs in the rhombohedral $R\bar{3}c$ structure and is not known to exhibit any structural or metal-insulator transitions, unlike other rare-earth nickelates. However, several experiments have hinted at the proximity of $LaNiO_3$ to other rare-earth nickelates. The high-temperature magnetic susceptibility, resistivity, and thermoelectric power of both $LaNiO_3$ and the orthorhombic rare-earth nickelates display similar features that are consistent with the presence of a heterogeneous phase consisting of two different Ni sites [25–27]. Optical and electron tunneling spectroscopy experiments show the presence of a pseudogap in thin films of both $LaNiO_3$ and $NdNiO_3$ [28, 29]. A pair density function analysis of the neutron scattering data of a powder $LaNiO_3$ sample by Li *et al.* found that the local structure is better described by monoclinic $P2_1/n$ and orthorhombic *Pnma* structures below and above 200 K, respectively [30].

The lack of high-quality single crystals has hindered a more rigorous determination of the properties of the rare-earth nickelates. Recently, two groups have reported the synthesis of $LaNiO_3$ single crystals using the floating zone technique under high oxygen pressures. Zhang *et al.*'s samples, which were grown under the oxygen pressure of 30–50 bar, were characterized to have the rhombohedral $R\bar{3}c$ structure and showed metallic conductivity [31]. These samples did not exhibit any structural or magnetic transition, but their magnetic susceptibility showed a broad maximum around 200 K. A small anomaly had also been observed around this temperature in earlier measurements of the magnetic susceptibility on polycrystalline samples [30]. However, the samples grown by Guo *et al.* under the oxygen pressure of 130–150 bar showed an antiferromagnetic transition at 157 K with an ordering wave vector of $(\frac{1}{4}, \frac{1}{4}, \frac{1}{4})_c$ in the pseudocubic notation [32]. The antiferromagnetic phase remained in the rhombohedral $R\bar{3}c$ structure and continued to exhibit a metallic behavior. This is rather surprising considering that the pair density function analysis mentioned above indicated a structural similarity at the nanoscale between $LaNiO_3$ and other rare-earth nickelates [30]. If an antiferromagnetic transition were to be present in $LaNiO_3$, one would have expected it to also show structural and metal-insulator transitions like other rare-earth nickelates.

The presence or absence of the breathing distortions in the antiferromagnetic phase of $LaNiO_3$ has important implications on the microscopic mechanism for the phase transition in the rare-earth nickelates. Lee *et al.* have suggested that Fermi surface nesting, not charge disproportionation, plays a key role in the phase transition of the rare-earth nickelates, especially the ones with larger rare-earth radii [33, 34]. They have shown that charge disproportionation necessarily occurs in the orthorhombic *Pnma* rare-earth nickelates as a secondary order parameter during the antiferromagnetic phase transition. On the other hand, their symmetry analysis within a Landau theory suggested that a pure antiferromagnetic state without any disproportionation occurs in the rhombohedral $R\bar{3}c$ phase. A lack of breathing distortions in antiferromagnetic $LaNiO_3$ would imply that the disproportionation suggested by Mazin *et al.* does not play a decisive role.

The above discussion amply demonstrates that the rhombohedral LaNiO$_3$ is close to the structural and magnetic phases that appear in other rare-earth nickelates. DFT calculations and its extensions DFT+$U$ and DFT+DMFT have been used to study the structural, electronic, and magnetic properties of the rare-earth nickelates [18–20, 23, 35–47]. Park *et al.* have claimed that DFT is inadequate to qualitatively describe the ground-state properties of the rare-earth nickelates and more sophisticated methods are necessary [42]. However, recent works by Varignon *et al.* [23] and Hampel and Ederer [45] show that rigorous DFT calculations can describe the antiferromagnetic and disproportionated ground state of these materials. These authors focused their studies on the orthorhombic rare-earth nickelates. A similar study on LaNiO$_3$ would be helpful in clarifying the structural, electronic, and magnetic properties of this material.

In this paper, I use DFT calculations to explore the structural, electronic, and magnetic instabilities of LaNiO$_3$. The calculated non-spin-polarized phonon dispersions of cubic LaNiO$_3$ show instabilities at the wave vectors $R\left(\frac{1}{2}, \frac{1}{2}, \frac{1}{2}\right)_c$ and $M\left(\frac{1}{2}, \frac{1}{2}, 0\right)_c$. I fully relaxed the supercells that exhibit the various Glazer tilts allowed by these instabilities. I find that several structures with different Glazer tilts lie close in energy. The orthorhombic *Pnma* structure is marginally lower in energy than the rhombohedral $R\bar{3}c$ structure in my calculations. This suggests the presence of structural fluctuations in LaNiO$_3$ at finite temperatures and indicates a possible proximity to a structural quantum critical point. Both structures exhibit the ↑0↓0 antiferromagnetic order with a rock-salt arrangement of the octahedral breathing distortions when the spin-polarized relaxations are performed. The gain in energy due to the breathing distortions is larger than the gain in energy due to the antiferromagnetic ordering, suggesting that the mechanism of disproportionation proposed by Mazin *et al.* plays a key role in the phase transition of LaNiO$_3$. These phases are semimetallic with small three-dimensional Fermi pockets. This is mostly consistent with the recent experiments of Guo *et al.* that uncovered an antiferromagnetic transition in LaNiO$_3$ without a concomitant metal-insulator transition [32]. They did not observe the breathing distortions that I find in my calculations, and this might be because the calculated difference of ∼0.01 Å between the Ni-O bond lengths of the compressed and expanded octahedra is very small.

## 2    Methods

The DFT calculations presented here were obtained using the pseudopotential-based planewave method as implemented in the QUANTUM ESPRESSO package [48]. The phonon dispersions were calculated using density functional perturbation theory [49]. The calculations were done within the generalized gradient approximation of Perdew, Burke and Ernzerhof (PBE GGA) [50] using the pseudopotentials generated by Garrity *et al.* [51]. Some calculations were also checked using the ONCV pseudopotentials [52], as well as Garrity *et al.*'s pseudopotentials within the local density approximation (LDA). The planewave basis-set and charge density expansions were done using cut-offs of 50 and 250 Ry, respectively.

I used a 16 × 16 × 16 $k$-point mesh for the Brillouin zone integration in the phonon calculations. The dynamical matrices were obtained on an 8 × 8 × 8 $q$-point grid which includes the special high-symmetry points $R\left(\frac{1}{2}, \frac{1}{2}, \frac{1}{2}\right)_c$ and $M\left(\frac{1}{2}, \frac{1}{2}, 0\right)_c$ emphasized below. The phonon dispersions were obtained by Fourier interpolation. The structural relaxation of the various Glazer tilts [53] were done on 40-atom 2 × 2 × 2 pseudocubic supercells using an 8 × 8 × 8 $k$-point mesh. I used denser meshes in the spin-polarized structural relaxations. For the 20- and 80-atom supercells of the $R\bar{3}c$ structure, I used 12 × 12 × 8 and 8 × 8 × 8 meshes, respectively. For the 40- and 80-atom supercells of the *Pnma* structure, 6 × 8 × 12 and 6 × 4 × 12 meshes were used, respectively. I made extensive use of the ISOTROPY [54] and SPGLIB [55] packages

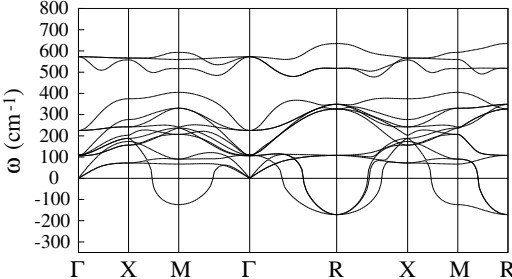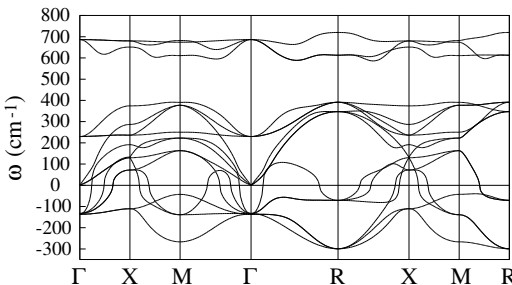

Figure 1: Calculated PBE GGA non-spin-polarized phonon dispersions of cubic LaNiO$_3$ (left) and cubic YNiO$_3$ (right). The imaginary frequencies are denoted by negative values.

in the symmetry analysis. VESTA [56] and XCRYSDEN [57] were used to visualize the crystal structures and Fermi surfaces, respectively.

To check convergence with respect to the planewave cut-off, I repeated the structural relaxations of the various Glazer tilts for a cut-off value of 60 Ry. Same energetic rankings were obtained. I also did some calculations with larger $k$-point meshes, and this did not change the results in a meaningful way. Note that the meshes used in this work are denser than those used in two recent DFT studies on the rare-earth nickelates [23, 45].

## 3 Results and Discussions

### 3.1 Non-spin-polarized structural relaxations

The rhombohedral $R\bar{3}c$ structure of LaNiO$_3$ is characterized by out-of-phase rotations of the oxygen octahedra about the three axes of the parent cubic phase and is denoted by $a^-a^-a^-$ in Glazer's notation. The orthorhombic $Pnma$ structure of all other rare-earth nickelates involves an in-phase rotation about one axis and out-of-phase rotations by a different amount about the two other axes and has the notation $a^+b^-b^-$. These distorted structures derive from different dynamical instabilities of the cubic perovskite phase. To examine if they have similar latent structural instabilities, I start by comparing the non-spin-polarized phonon dispersions of cubic LaNiO$_3$ and, as a representative member of the orthorhombic family, cubic YNiO$_3$, which are shown in Figs. 1(a) and (b), respectively. These were calculated within the PBE GGA using the relaxed lattice parameters of $a = 3.837$ and 3.745 Å for LaNiO$_3$ and YNiO$_3$, respectively.

The phonon dispersions of the two nickelates in the cubic phase show several branches that are unstable along different directions in the Brillouin zone. The phonon instabilities are weaker in LaNiO$_3$ than in YNiO$_3$, consistent with the observation that LaNiO$_3$ is closer to the cubic phase. In both materials, the largest instability occurs at the wave vector $R$ $(\frac{1}{2}, \frac{1}{2}, \frac{1}{2})_c$ in the pseudocubic notation. This mode is triply degenerate and has the irreducible representation (irrep) $R_4^+$ when the convention that Ni is placed at $(0, 0, 0)$ is used. [Its irrep is $R_5^-$ when Ni is at $(0.5, 0.5, 0.5)$.] Another mode at $M$ $(\frac{1}{2}, \frac{1}{2}, 0)_c$ also shows a large instability. It is singly degenerate and has the irrep $M_3^+$ when Ni is placed at $(0, 0, 0)$. The $R_4^+$ mode induces out-of-phase rotations of the oxygen octahedra about the axes, while the $M_3^+$ mode generates in-phase rotations. Note that although the unstable phonon at $M$ is singly degenerate, the star of $M$ has three points. So the $M_3^+$ mode is able to generate in-phase rotations about all three axes.

The $a^-a^-a^-$ tilt system of the rhombohedral LaNiO$_3$ requires freezing of the distortions due to only the $R_4^+$ mode. On the other hand, because of the presence of an in-phase rotation

Table 1: The relative total energies of LaNiO$_3$ and YNiO$_3$ with different Glazer tilts. The energies of the Glazer tilts that could not be stabilized are denoted by "—".

| tilt system | space group | LaNiO$_3$ energy (meV/Ni) | YNiO$_3$ energy (meV/Ni) |
|---|---|---|---|
| $a^0a^0a^0$ | $Pm\bar{3}m$ | 0.0 | 0.0 |
| $a^+a^+a^+$ | $Im\bar{3}$ | −29.6 | −461.3 |
| $a^0b^+b^+$ | $I4/mmm$ | −33.3 | −489.7 |
| $a^0a^0c^+$ | $P4/mbm$ | −45.1 | −491.0 |
| $a^0a^0c^-$ | $I4/mcm$ | −107.9 | −613.4 |
| $a^0b^-b^-$ | $Imma$ | −114.6 | −788.6 |
| $a^-a^-a^-$ | $R\bar{3}c$ | −115.9 | −767.3 |
| $a^+b^+c^+$ | $Immm$ | — | — |
| $a^+a^+c^-$ | $P4_2/nmc$ | — | −715.2 |
| $a^0b^+c^-$ | $Cmcm$ | — | −731.3 |
| $a^+b^-b^-$ | $Pnma$ | −116.9 | −1010.1 |
| $a^0b^-c^-$ | $C2/m$ | — | — |
| $a^-b^-b^-$ | $C2/c$ | — | −767.5 |
| $a^+b^-c^-$ | $P2_1/m$ | — | — |
| $a^-b^-c^-$ | $P\bar{1}$ | — | −788.6 |

along one axis, the $a^+b^-b^-$ tilt system favored by the orthorhombic YNiO$_3$ requires freezing of the distortions due to both $R_4^+$ and $M_3^+$ modes. The value of the imaginary frequency of the $R_4^+$ mode $\omega_{R_4^+} = 171i$ cm$^{-1}$ is noticeably larger than that of the $M_3^+$ mode $\omega_{M_3^+} = 124i$ cm$^{-1}$ in LaNiO$_3$, indicating that the distortions due to the $R_4^+$ mode might be more favorable in this material. However, the instabilities of the $R_4^+$ and $M_3^+$ modes, with the respective frequencies $\omega_{R_4^+} = 299i$ and $\omega_{M_3^+} = 267i$ cm$^{-1}$, are much closer in YNiO$_3$, suggesting that distortions due to both modes are likely to occur in YNiO$_3$. Thus the calculated phonon instabilities seemingly provide the microscopic explanation for the different octahedral rotations observed in LaNiO$_3$ and YNiO$_3$.

Although the calculated phonon instabilities are consistent with the observed structural distortions in the two nickelates, these instabilities could also lead to other structural distortions. For example, phonon instabilities similar to that of LaNiO$_3$ occur in SrTiO$_3$, but they cause out-of-phase rotations of the oxygen octahedra about only one axis ($a^0a^0c^-$ in Glazer's notation) in SrTiO$_3$ [58]. Howard and Stokes have shown that fifteen different structures can arise out of the $R_4^+$ and $M_3^+$ phonon instabilities [59]. I generated all fifteen structures for both LaNiO$_3$ and YNiO$_3$ and fully relaxed them using DFT calculations within the PBE GGA. The energies of the relaxed structures relative to that of the undistorted structure are given in Table 1.

The calculations show that the gain in energy due to octahedral rotations in LaNiO$_3$ is relatively small compared to that in YNiO$_3$, which again confirms that LaNiO$_3$ is close to the cubic phase. Not all octahedral tilt patterns could be stabilized, and these structures are denoted by the symbol "—" in the table. The energy of the $R\bar{3}c$ structure of LaNiO$_3$ with the tilt pattern $a^-a^-a^-$ is −115.9 meV/Ni (i.e, per formula unit that consists of five atoms) relative to that of the undistorted cubic structure. Surprisingly, I find that the energy of the $Pnma$ structure with the tilt pattern $a^+b^-b^-$ to be even lower, albeit by only 1.0 meV/Ni. In addition, the structure with the tilt pattern $a^0b^-b^-$ is only 1.3 meV/Ni higher in energy than the $R\bar{3}c$ phase.

The closeness in energy of several distinct structures of LaNiO$_3$ indicates that the structure

of this material can dynamically fluctuate at finite temperatures and suggests that the material might be in the proximity of a structural quantum critical point.

All known diffraction experiments on powder and single crystal samples of LaNiO$_3$ have found the structure to be rhombohedral with the space group $R\bar{3}c$, but I find the orthorhombic structure with the space group $Pnma$ to be lower in energy. To check the robustness of my calculations, I also did structural relaxations using other sets of pseudopotentials (GRVB LDA and ONCV PBE), and they also give the lowest energy to the $Pnma$ phase. The only experiment that is consistent with my finding is the pair density function analysis of powder LaNiO$_3$ performed by Li *et al.* who found that the high-temperature phase of LaNiO$_3$ is best described by an orthorhombic $Pnma$ structure at the nanoscale [30].

For YNiO$_3$, the orthorhombic $Pnma$ structure with the $a^+b^-b^-$ tilt pattern has the lowest energy. The energies of other tilt patterns of YNiO$_3$ are much higher than the $Pnma$ structure, unlike in the case of LaNiO$_3$. For example, the $Imma$ structure with the $a^0b^-b^-$ tilt pattern, which is energetically closest to the $Pnma$ structure, is higher in energy by 221.5 meV/Ni. As noted above, the imaginary frequencies of the $R_4^+$ and $M_3^+$ modes are closer in YNiO$_3$ than in LaNiO$_3$. So the presence of competing structural phases and possible proximity to a structural quantum critical point is not caused by a near degeneracy of the $R_4^+$ and $M_3^+$ phonon instabilities. The results shown here suggest that a larger difference between the imaginary frequencies of the two phonon modes might lead to such a competition, although a very large difference would probably stabilize a structural distortion due to only one unstable mode.

## 3.2 Spin-polarized structural relaxations

Guo *et al.* have recently reported an antiferromagnetic phase transition at $T_N \sim 157$ K in their single crystal samples of LaNiO$_3$ [32]. The propagation wave vector observed for the antiferromagnetic ordering in LaNiO$_3$ is $(\frac{1}{4}, \frac{1}{4}, \frac{1}{4})_c$ in the cubic notation. In terms of the reciprocal lattice vectors of the rhombohedral $R\bar{3}c$ and orthorhombic $Pnma$ unit cells, the propagating wave vectors are $(\frac{1}{2}, \frac{1}{2}, \frac{1}{2})_r$ and $(\frac{1}{2}, \frac{1}{2}, 0)_o$, respectively.

Another recent experimental study on single crystal samples by Zhang *et al.*, however, did not find any such transition [31]. A broad feature in the susceptibility measurements is seen in Zhang *et al.*'s samples and a small anomaly had also been observed previously [30], which suggests that long range magnetic ordering might only occur in highly pure samples.

To understand the nature of magnetic instabilities, if there are any, in LaNiO$_3$ and possible competition between different magnetic interactions, I extensively studied the stability of diverse magnetically ordered phases in several supercells of $R\bar{3}c$ and $Pnma$ structures using spin-polarized DFT calculations within the PBE GGA.

### 3.2.1 Spin-polarized structural relaxations in $R\bar{3}c$ LaNiO$_3$

In $R\bar{3}c$ LaNiO$_3$, I found a weak ferromagnetic instability with an ordered moment of 0.2 $\mu_B$/Ni and an energy gain of 0.3 meV/Ni relative to the paramagnetic state. I was not able to stabilize the $A$-, $C$-, or $G$-type orderings in a 40-atom supercell of the $R\bar{3}c$ structure. All these orderings showed negligible moments and no discernible gain in energy.

The experimentally observed antiferromagnetic order corresponds to an 80-atom $2 \times 2 \times 2$ supercell of the $R\bar{3}c$ structure. Since the $R\bar{3}c$ unit cell has two formula units, there are sixteen Ni atoms in this supercell. The propagation vector uniquely determines the ordering pattern of eight of these Ni atoms. The ordering pattern in the sublattice formed by the remaining eight atoms is also uniquely determined by the propagation vector, and each Ni atom in one sublattice has three spin-up and three spin-down Ni sites of the other sublattice as its nearest neighbors. So there is only one collinear antiferromagnetic ordering pattern consistent with the reported ordering wave vector in this $R\bar{3}c$-derived supercell. However, the two sublattices

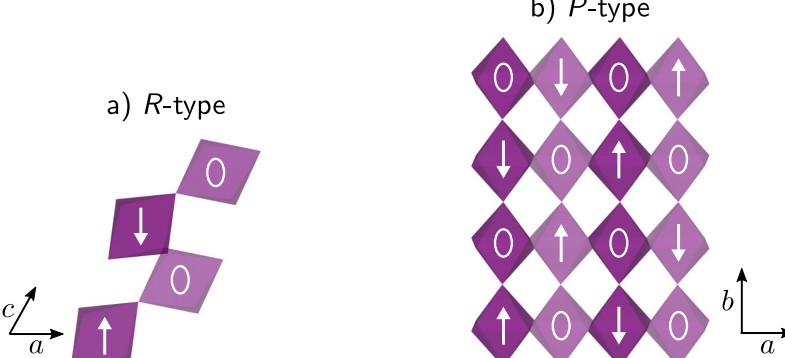

Figure 2: The a) $R$- and b) $P$-type phases derived from the $R\bar{3}c$ and $Pnma$ structures with the propagation wave vectors $(0, 0, \frac{1}{2})_r$ and $(\frac{1}{2}, \frac{1}{2}, 0)_o$, respectively. Only the smallest units of repetition are shown. The octahedra in different layers in the out-of-plane direction are shaded differently. The octahedra with the symbol "0" have smaller volumes, and the Ni sites inside them have no magnetic moments.

have the freedom to have different values for the on-site magnetic moments. In case such an antiferromagnetic state with different on-site moment occurs, it would correspond to a rock-salt ordering of the breathing distortions in the rhombohedral phase similar to the one observed for the orthorhombic rare-earth nickelates. However, I was not able to stabilize any magnetic orderings with noticeable magnetic moments and energy gains in this 80-atom unit cell.

I also tried to stabilize antiferromagnetic orderings in the 20-atom $1 \times 1 \times 2$ supercell corresponding to the propagation wave vector $(0, 0, \frac{1}{2})_r$ of the $R\bar{3}c$ unit cell. This wave vector is equivalent to $(\frac{1}{4}, \frac{1}{4}, \frac{3}{4})_c$ in the pseudocubic notation, and it is also compatible with the neutron scattering experiments of Guo *et al.* [32]. In this supercell, the ordering wave vector uniquely determines the collinear antiferromagnetic ordering pattern up to any disproportion-ation. The wave vector partitions the four Ni atoms in this supercell into two sublattices with two Ni each, and each Ni site in one sublattice has three spin-up and three spin-down Ni sites from the other sublattice as its nearest neighbors. The spins are ordered ferromagnetically in the $ab$ plane and antiferromagnetically along the $(0, 0, 1)_r$ direction. I was able to obtain such an antiferromagnetic solution with an energy gain of 0.7 meV/Ni relative to the nonmag-netic state. The two Ni sublattices have slightly different magnitudes of 0.27 and 0.26 $\mu_B$ for the magnetic moments, so this phase already shows a tendency towards disproportionation. The disproportionated state shows an even larger energy gain of 2.1 meV/Ni relative to the nonmagnetic phase and has Ni sites with magnetic moments of 0.6 $\mu_B$ and zero inside the larger and smaller octahedra, respectively. The disproportionated octahedra are arranged in a rock-salt-type arrangement, consistent with Mazin *et al.*'s picture of charge ordering [18].

The presence of the breathing distortions in the $R\bar{3}c$-derived supercell is, however, incon-sistent with Lee *et al.*'s symmetry analysis within a Landau theory, which suggested that no charge ordering occurs in the antiferromagnetic ordering of the rhombohedral structure [33].

Although the difference in the magnitude of the magnetic moments is large in the dispro-portionated phase, the breathing distortions are relatively small. The larger octahedra have a volume of 10.1 Å$^3$, while the smaller ones have a volume of 9.8 Å$^3$. The Ni-O bond lengths in the two sets of octahedra are 1.95 and 1.96 Å, respectively, and this small difference of 0.01 Å might be the reason for the difficulty in observing this distortion in the experiments. This disproportionated antiferromagnetic state is labeled as $R$-type in this work and is illustrated in Fig. 2(a).

### 3.2.2 Spin-polarized structural relaxations in *Pnma* LaNiO$_3$

Since the orthorhombic *Pnma* phase of LaNiO$_3$ has a slightly lower energy than the $R\overline{3}c$ phase in my calculations, I also explored if antiferromagnetism occurs in the orthorhombic phase. I was able to stabilize a ferromagnetic state with an ordered moment of 0.2 $\mu_B$/Ni and an energy gain of 0.6 meV/Ni relative to the nonmagnetic phase. But I was not able to stabilize the *A*-, *C*-, or *G*-type ordering arrangements.

I constructed an 80-atom $2 \times 2 \times 1$ supercell of the *Pnma* structure that corresponds to the experimentally observed propagation wave vector of the antiferromagnetic order. The *Pnma* unit cell has four formula units, so this supercell has sixteen Ni atoms. The constraint due to the propagation wave vector partitions the Ni lattice into four sublattices with four Ni atoms each. One can enumerate eight arrangements of collinear spin orderings within this constraint, but only two of them are symmetrically inequivalent. These are the so-called *S*- and *T*-type orderings [36]. In both these orderings, the spins are ordered ↑↑↓↓ in the *ac* plane and are flipped to ↓↓↑↑ in the next-nearest plane along the *b* direction. They are sandwiched by nearest-neighbor planes with the same types of spin arrangements in the *S*-type ordering, while in the *T*-type ordering, they are sandwiched by layers with the spin orderings ↑↓↓↑ and ↓↑↑↓. Like the spin orderings in the rhombohedral supercells that are compatible with the experimentally observed propagating wave vector, each Ni atom in these spin orderings have three spin-up and spin-down Ni atoms as their nearest neighbors. In addition, the four Ni sublattices in the supercell have the freedom to disproportionate and have different on-site moments.

I was able to stabilize both the *S*- and *T*-type antiferromagnetic orderings in the $2 \times 2 \times 1$ supercell of the orthorhombic LaNiO$_3$. The energy gain of ∼0.4 meV/Ni relative to the nonmagnetic phase due to these orderings is small, like in the nondisproportionated antiferromagnetic state corresponding to the wave vector $(0, 0, \frac{1}{2})_r$ of the $R\overline{3}c$ unit cell. The on-site Ni moments are ∼0.2 $\mu_B$ in the *S*- and *T*-type antiferromagnetic phases, but the magnitudes of the moments vary by ∼5% in different Ni sublattices. I found that both these orderings show a strong propensity to disproportionate in a rock-salt pattern of alternating large and small NiO$_6$ octahedra. The Ni sites inside the large octahedra have a moment of 0.6 $\mu_B$, while the ones inside the small octahedra have no magnetic moment. The disproportionated phases of the *S*- and *T*-type orderings are symmetrically identical, and this phase has an energy gain of 2.0 meV/Ni relative to the nonmagnetic phase. This phase is labeled as *P*-type and is shown in Fig. 2(b). The larger and smaller octahedra in this structure have volumes of 10.2 and 9.9 Å$^3$, respectively. The Ni-O distances in the corresponding octahedra are 1.97 and 1.95 Å. These values are similar to the one obtained for the $R\overline{3}c$-derived *R*-type phase. In fact, the *P*- and *R*-type phases are only distinguished by their underlying crystal structures. The magnetic ordering is same in these phases, with the Ni sites with zero moments having three spin-up and three spin-down nearest neighbors.

## 3.3 Lindhard susceptibility and antiferromagnetic ordering

All three nondisproportionated antiferromagnetic solutions that I obtained for LaNiO$_3$ exhibit a propensity for the octahedral breathing distortions. This supports Mazin *et al.*'s theory that the nickelates occur in a crossover between the localized and itinerant regimes where the nearest-neighbor Ni sites have a tendency to disproportionate [18]. However, the 80-atom supercell of the $R\overline{3}c$ structure corresponding to the propagation vector $(\frac{1}{2}, \frac{1}{2}, \frac{1}{2})_r$ also allows simultaneous existence of both antiferromagnetism and rock-salt-type disproportionation. But I could not stabilize this phase. To determine if there is any connection between the Fermi

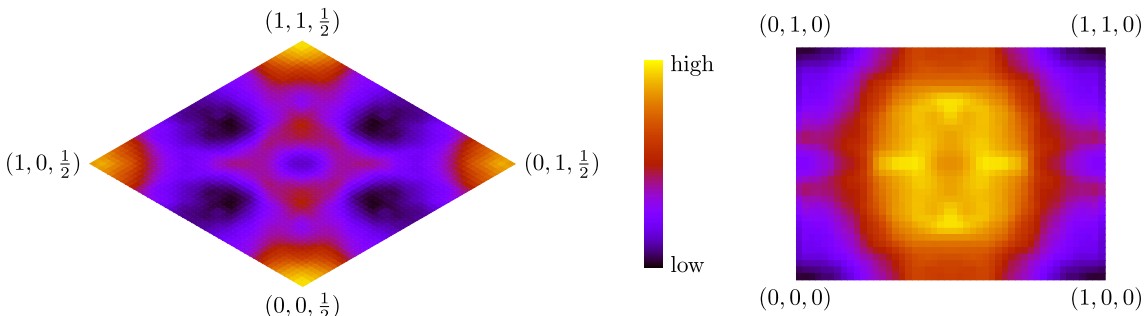

Figure 3: Calculated Lindhard susceptibility of LaNiO$_3$ in the $R\overline{3}c$ (left) and $Pnma$ (right) structures.

surface instabilities and the magnetic orderings, I calculated the Lindhard susceptibility

$$\chi_0(q,\omega) = \sum_{k,m,n} |M^{m,n}_{k,k+q}|^2 \frac{f(\epsilon^m_k) - f(\epsilon^n_{k+q})}{\epsilon^m_k - \epsilon^n_{k+q} - \omega - \iota\delta}$$

at $\omega \to 0$ and $\delta \to 0$, where $\epsilon^m_k$ is the energy of a band $m$ at the wave vector $k$ and $f$ is the Fermi distribution function. $M$ is the matrix element, which is set to unity for the constant matrix element approximation employed here. The negligence of the matrix element changes the relative intensities of the peaks in the susceptibility [60]. However, major features remain the same and qualitative understanding can still be gleaned off from such an approximation. I note that a previous discussion of the magnetic susceptibility in the nickelates has also made this approximation [34].

The calculated Lindhard susceptibility of the $R\overline{3}c$ phase for the $(q_a, q_b, \frac{1}{2})_r$ plane is shown in Fig. 3(a). The susceptibility shows a peak at $(0,0,\frac{1}{2})_r$, the propagation wave vector for which I was able to stabilize an antiferromagnetic solution. The calculated susceptibility at $(\frac{1}{2}, \frac{1}{2}, \frac{1}{2})_r$ is low and occurs at a local minima, and this seems to explain the lack of antiferromagnetic instability at this wave vector in the calculations. This result is consistent with Lee *et al.*'s suggestion that Fermi surface nesting plays an important role in the antiferromagnetic instability of the rare-earth nickelates [33,34]. [But note the discussion in the following paragraph.] However, it is the nesting instability of the rhombohedral, not the cubic, phase that is important because a peak at the wave vector $(\frac{1}{4}, \frac{1}{4}, \frac{1}{4})_c = (\frac{1}{2}, \frac{1}{2}, \frac{1}{2})_r$ does occur in the cubic phase [33,34].

The calculated Lindhard susceptibility of the orthorhombic $Pnma$ LaNiO$_3$, which is shown in Fig. 3(b), does not exhibit a sharp peak at $(\frac{1}{2}, \frac{1}{2}, 0)_o$ that corresponds to the experimentally observed ordering wave vector. Instead of sharp peaks, the susceptibility of the $Pnma$ phase shows a plateau-like enhancement of the susceptibility in the region $(\frac{1}{4} < q_a < \frac{3}{4}, 0 < q_b < 1, 0)_o$. The wave vector $(\frac{1}{2}, \frac{1}{2}, 0)_o$ actually occurs at a local minimum. Therefore, although a high Lindhard susceptibility appears necessary for the disproportionated antiferromagnetic instabilities, a sharp peak corresponding to a well-defined nesting does not seem to be crucial. Importantly, because there are four Ni sublattices, the antiferromagnetic orderings in the 80-atom supercell corresponding to the wave vector $(\frac{1}{2}, \frac{1}{2}, 0)_o$ can host either the nearest-neighbor rock-salt-type disproportionation or the layered disproportionation where the neighboring Ni planes are alternatingly disproportionated. However, I was not able to stabilize the latter arrangement of disproportionation, which shows that the nearest-neighbor rock-salt ordering of the breathing distortions play an essential role in the phase transition of LaNiO$_3$.

The sizable energy gain due to rock-salt-type disproportionations, and the lack of such a gain in other arrangements of the breathing distortions, indicates that the structural instability that leads to the breathing distortions is not just related to the moment formation. In addition

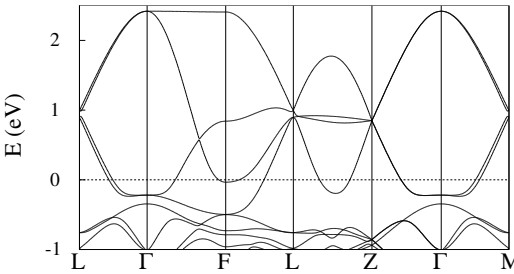
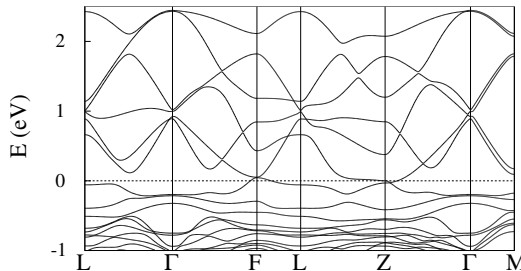

Figure 4: Calculated band structures of the nonmagnetic $R\bar{3}c$ LaNiO$_3$ (left) and the disproportionated antiferromagnetic $R$-type LaNiO$_3$ corresponding to the propagation wave vector $(0,0,\frac{1}{2})_r$. The band structures are plotted along the path $L$ $(\frac{1}{2},0,0)_r \rightarrow \Gamma$ $(0,0,0) \rightarrow F$ $(\frac{1}{2},\frac{1}{2},0)_r \rightarrow L$ $(\frac{1}{2},0,0)_r \rightarrow Z$ $(\frac{1}{2},\frac{1}{2},\frac{1}{2})_r \rightarrow \Gamma$ $(0,0,0)$ $\rightarrow M$ $(0,0,\frac{1}{2})_r$. The coordinates are given in terms of the reciprocal lattice vectors of the primitive cell.

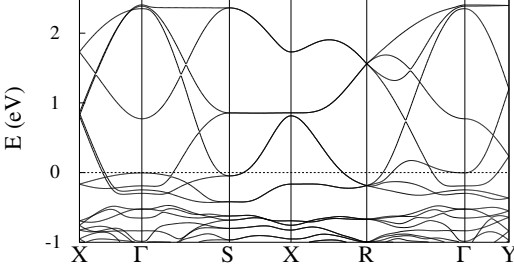
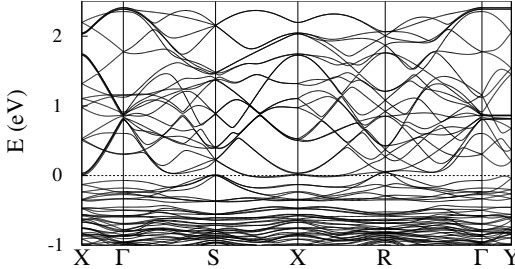

Figure 5: Calculated band structures of the nonmagnetic $Pnma$ LaNiO$_3$ (left) and the disproportionated antiferromagnetic $P$-type LaNiO$_3$ corresponding to the propagation wave vector $(\frac{1}{2},\frac{1}{2},0)_o$. The band structures are plotted along the path $X$ $(\frac{1}{2},0,0)_o \rightarrow \Gamma$ $(0,0,0) \rightarrow S$ $(\frac{1}{2},0,\frac{1}{2})_o \rightarrow X$ $(\frac{1}{2},0,0)_o \rightarrow R$ $(\frac{1}{2},\frac{1}{2},\frac{1}{2})_o \rightarrow \Gamma$ $(0,0,0)$ $\rightarrow Y$ $(0,\frac{1}{2},0)_o$. The coordinates are given in terms of the reciprocal lattice vectors of the primitive cell.

to the moment formation, my calculations suggest that the breathing instability arises due to coupling with the antiferromagnetic fluctuations that involve three spin-up and spin-down nearest neighbors. This is also supported by the fact that I was not able to stabilize breathing distortions in the non-spin-polarized calculations.

## 3.4 Electronic structure of the disproportionated antiferromagnetic phases

The calculated band structures of LaNiO$_3$ in the nonmagnetic $R\bar{3}c$ and $Pnma$ structures without the disproportionation and in the corresponding low-symmetry $R$- and $P$-type antiferromagnetic phases with the breathing distortions are shown in Figs. 4 and 5, respectively. For comparison, the band structures of YNiO$_3$ in the nonmagnetic and $P$-type phases are also shown in Fig. 6.

From the band structures of the nonmagnetic phases, one can readily identify the antibonding $e_g$-derived bands between $-0.5$ and $2.5$ eV relative to the Fermi level. There are two spin-degenerate $e_g$ bands per Ni in each structures. For example, the $R\bar{3}c$ structure with two Ni per primitive cell has four bands in this manifold [Fig. 4(left)], while the $Pnma$ structure with four Ni per primitive cell has eight bands [Fig. 5(left)]. This $e_g$-derived manifold is completely

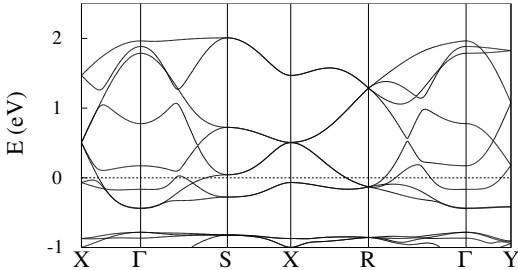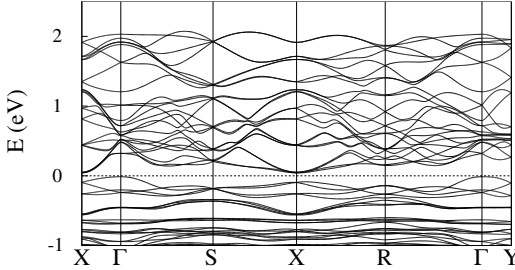

Figure 6: Calculated band structures of the nonmagnetic *Pnma* YNiO$_3$ (left) and the disproportionated antiferromagnetic *P*-type YNiO$_3$ corresponding to the propagation wave vector $(\frac{1}{2}, \frac{1}{2}, 0)_o$. The path is as in Fig. 5.

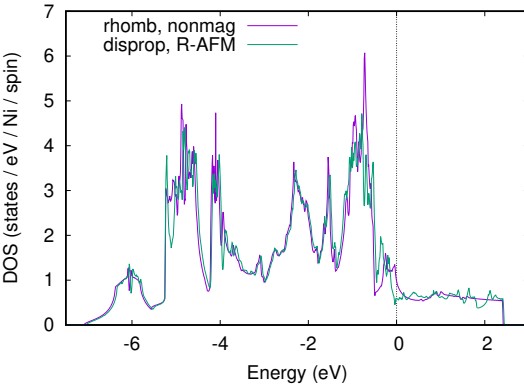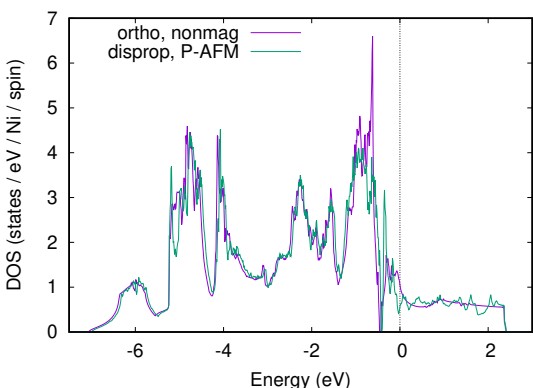

Figure 7: Left: Calculated electronic DOS of LaNiO$_3$ in the rhombohedral nonmagnetic and disproportionated antiferromagnetic *R*-type structures. Right: DOS of LaNiO$_3$ in the orthorhombic non-magnetic and disproportionated antiferromagnetic *P*-type structures.

separated in the *Pnma* structure, but it touches a lower manifold in the $R\bar{3}c$ structure. [Such a crossing is allowed in the $R\bar{3}c$ structure because the bands belonging to the lower manifold also have the $e_g$ representation in this space group.] The Fermi level lies near the bottom of this manifold and corresponds to a quarter filling. The Fermi surfaces in these phases are large and consist of multiple sheets (see Figs. 10 and 11 in the appendix). The electronic density of states (DOS) at the Fermi level is 1.0 states eV$^{-1}$ per Ni per spin in both the $R\bar{3}c$ and *Pnma* structures. This corresponds to a Sommerfeld coefficient of $\gamma = 4.7$ mJ mol$^{-1}$ K$^{-2}$, which is about 3.6 times smaller than the experimentally determined value of 17 mJ mol$^{-1}$ K$^{-2}$ [32,61].

As can be seen in the right panels of Figs. 4 and 5, the antiferromagnetic ordering and breathing distortions in LaNiO$_3$ have a drastic effect only on the lower half of the $e_g$-derived manifold. This is also apparent from the DOS plot shown in Fig. 7 that reveals a large decrease of the DOS value at the Fermi energy due to a downward movement of a peak in the low-symmetry phases.

In the nonmagnetic phases, the lower and upper halves of the $e_g$-derived manifold touch at isolated points in the Brillouin zone. If one considers the two halves of the $e_g$-derived manifold as either separate or weakly coupled to each other, the lower half with one $e_g$ band per Ni is nominally half filled, and the antiferromagnetic ordering splits apart this half-filled manifold. For example, Fig. 4(right) shows the band structure of the disproportionated antiferromagnetic *R*-type ordered phase of LaNiO$_3$, which occurs in a $1 \times 1 \times 2$ supercell of the nonmagnetic phase corresponding to the propagation wave vector $(0, 0, \frac{1}{2})_r$. The number of bands are now

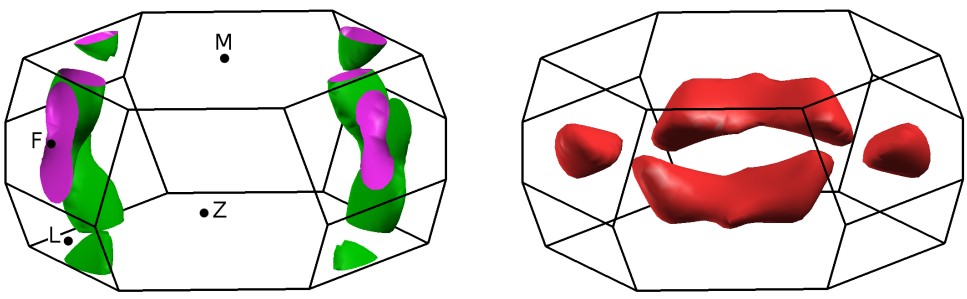

Figure 8: Left: Calculated Fermi sheets of LaNiO$_3$ in the disproportionated antiferromagnetic $R$-type phase.

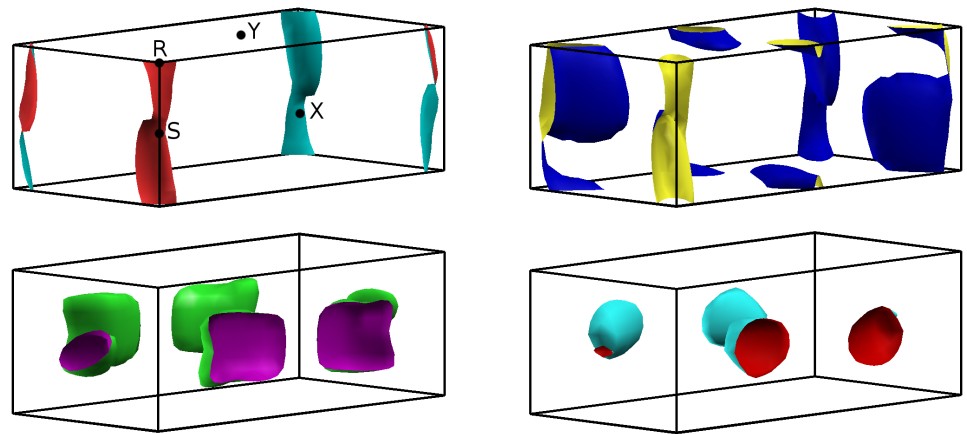

Figure 9: Left: Calculated Fermi sheets of LaNiO$_3$ in the disproportionated antiferromagnetic $P$-type phase.

doubled to eight in the $e_g$-derived manifold because there are four Ni atoms in this supercell. Between ∼1 and 2.5 eV, i.e. in the upper half of the $e_g$-derived manifold, the transition to the antiferromagnetic and disproportionated phase does not induce any dramatic gaps. The four bands in this upper half develop small gaps to remove the degeneracies that arise from band foldings, but these four bands largely exhibit the imprint of the two highest bands of the $R\bar{3}c$ structure. However, a large rupture appears in the lower half of the $e_g$-derived manifold. Two bands are shifted above the Fermi level and two bands below it, and the electronic structure near the Fermi level looks nothing like that of the nonmagnetic phase without the disproportionation.

The Fermi surfaces of the disproportionated antiferromagnetic $R$- and $P$-type phases, which are shown in Figs. 8 and 9, respectively, also illustrate this dramatic change. It is remarkable that the band structure, DOS, and Fermi surface change greatly due to the ↑0↓0-type antiferromagnetic ordering and breathing distortions even though the total energies of the high- and low-symmetry phases differ by only ∼2.0 meV/Ni. This indicates a strong coupling between the electrons at the Fermi energy, ↑0↓0-type antiferromagnetic order, and breathing distortions. The longitudinal magnetic fluctuations associated with this coupling might damp the magnitude of the moments at the Ni sites and may explain the small, so far undetermined value of the ordered moments in LaNiO$_3$ [32].

The disproportionated antiferromagnetic phases of LaNiO$_3$ are semimetallic and have band crossings generating small Fermi pockets. In YNiO$_3$, however, a gap appears due to a complete splitting of the lower half of the $e_g$-derived manifold.

The Fermi pockets in the disproportionated antiferromagnetic phases of LaNiO$_3$ are highly

anisotropic. As can be seen in the right panels of Figs. 4 and 5, these result from the Fermi level crossing of the bands that are highly dispersive along certain directions and flat along others. Furthermore, the band crossings occur near the edges and faces of the Brillouin zone, which leads to the presence of valley degeneracies. These characteristics cause the densities of states at the Fermi level to be relatively large in the low-symmetry phases even though the Fermi pockets enclose small portions of the Brillouin zone, yielding small carrier concentrations. I obtain DOS values of 0.58 and 0.56 states eV$^{-1}$ per Ni per spin for the $R$-type and $P$-type phases, respectively. This yields a calculated Sommerfeld coefficient of $\gamma \sim 2.7$ mJ mol$^{-1}$ K$^{-2}$, which is a reduction of around 40% compared to the nondisproportionated, nonmagnetic phases.

The semimetallic electronic structure that I have obtained here for the disproportionated antiferromagnetic phases agrees with the results of Guo *et al.* who find a metallic conductivity also in the low-temperature antiferromagnetic phase of LaNiO$_3$ [32]. In their electrical resistivity measurements, they find that the resistivity of LaNiO$_3$ decreases even more rapidly below the antiferromagnetic transition. This suggests a suppression of the scattering channels below the transition temperature. Such a behavior is also found in LaFeAsO, where the resistivity decreases at a faster rate below the antiferromagnetic and structural phase transition [62, 63]. What is incredible about LaNiO$_3$ is that its resistivity is in the $\mu\Omega$ cm range in Guo *et al.*'s measurements. This is three orders of magnitude lower than that of LaFeAsO, which has resistivity in the m$\Omega$ cm range [62, 63]. The presence of highly dispersive band crossings at the Fermi level in both the nondisproportionated, nonmagnetic and disproportionated, antiferromagnetic phases might underlie this behavior.

Another striking feature of Guo *et al.*'s measurments is the large value of 17 mJ mol$^{-1}$ K$^{-2}$ obtained for the Sommerfeld coefficient $\gamma$, which is around five times larger than the value calculated here for the disproportionated antiferromagnetic phases. Previous ARPES [44, 64–66], optical conductivity [28, 67, 68], and thermodynamics measurements [61] have also identified large electron mass enhancement and possible formation of pseudogapped state in LaNiO$_3$, and this has been discussed in terms of strong local electronic correlations. When the values for the on-site Coulomb $U$ and Hund's rule $J$ are used such that they reproduce the experimentally measured electronic structure, DMFT calculations can explain such an enhancement [44, 66]. The calculations presented here show that different structural phases occur close in energy in both the nonmagnetic and antiferromagnetic LaNiO$_3$, and the nonlocal fluctuations between these structures might provide an additional avenue for the enhancement. The enhancement could also occur due to the longitudinal magnetic fluctuations arising out of the strong coupling between the electrons near the Fermi level, breathing distortions, and ↑0↓0 antiferromagnetic order. In particular, the pseudogap features observed in LaNiO$_3$ are likely the result of the changes in the electronic structure caused by the inchoate disproportionated antiferromagnetism present in this material.

## 4 Summary and Conclusions

This work was motivated by three recent experimental studies on LaNiO$_3$. i) The pair density analysis of the neutron scattering data on a powder sample by Li *et al.*, which showed that the nanoscale structure of LaNiO$_3$ can be best described by the *Pnma* and *P*2$_1$/*n* structures above and below 200 K, respectively [30]. ii) The x-ray diffraction, transport, and thermodynamic experiments on single crystal samples by Zhang *et al.* that showed the material to be rhombohedral, metallic, and paramagnetic down to 1.8 K [31]. iii) The neutron scattering, transport, and thermodynamic experiments on single crystal samples by Guo *et al.* that showed an antiferromagnetic transition at 157 K but no structural and metal-insulator transitions [32]. I used DFT calculations to explore the structural, electronic, and magnetic instabilities in

LaNiO$_3$ indicated by these experiments. The non-spin-polarized phonon dispersions of cubic LaNiO$_3$ show instabilites at the wave vectors $R$ $(\frac{1}{2}, \frac{1}{2}, \frac{1}{2})_c$ and $M$ $(\frac{1}{2}, \frac{1}{2}, 0)_c$ in the pseudocubic notation. I relaxed different supercells with all possible Glazer tilts allowed by these instabilities and found that several structures lie close in energy. In my calculations, the $Pnma$ phase is marginally lower in energy than the $R\bar{3}c$ phase. This suggests the presence of structural fluctuations at finite temperatures and a possible proximity to a structural quantum critical point. I was able to stabilize several ↑↑↓↓ antiferromagnetic configurations consistent with the experimentally observed wave vector $(\frac{1}{4}, \frac{1}{4}, \frac{1}{4})_c$ in both structural phases. These occur in 20-atom $1 \times 1 \times 2$ and 80-atom $2 \times 2 \times 1$ supercells of the $R\bar{3}c$ and $Pnma$ structures, respectively. In both these structures, the antiferromagnetic ordering caused an energy gain of only 0.7–0.4 meV/Ni relative to the respective nonmagnetic phases. The magnetic moment per Ni in these configurations is 0.2 $\mu_B$.

The antiferromagnetic states are highly susceptible to octahedral breathing distortions with a rock-salt ordering. Both phases relaxed to the disproportionated ↑0↓0 state with moments of 0.6 $\mu_B$ and zero at the Ni sites inside the large and small oxygen octahedra, respectively. The energies of both the $R\bar{3}c$- and $Pnma$-derived disproportionated antiferromagnetic phases are around 2.0 meV/Ni lower than the respective nonmagnetic phases. The larger energy gain due to the breathing distortions indicate that the disproportionation plays a key role in the phase transition of the rare-earth nickelates, as suggested by Mazin *et al.* [18]. The appearance of the breathing distortions in the $R\bar{3}c$-derived phase is at variance with an earlier theoretical work, which suggested that the magnetism in the $R\bar{3}c$ phase will occur without disproportionation [33].

The disproportionated antiferromagnetic phases derived from both the $R\bar{3}c$ and $Pnma$ structures are semimetallic with small three-dimensional Fermi pockets. This is consistent with the recent results of Guo *et al.* who observed an antiferromagnetic transition in LaNiO$_3$ without a concomitant metal-insulator transition [32]. They did not observe the structural distortions that I find, perhaps because the difference between the Ni-O bond lengths in the large and small octahedra is only about 0.01–0.02 Å. The transition to the disproportionated antiferromagnetic phases causes a large change in the electronic structure, and this may explain the observation of a pseudogap in the earlier optical and tunneling spectroscopy experiments. In addition, the structural and longitudinal magnetic fluctuations suggested by these calculations may provide an explanation for the high electron mass enhancement observed in this material.

## Acknowledgements

I am grateful to Benoît Fauqué for helpful discussions. This work was supported by the European Research Council grant ERC-319286 QMAC and the Swiss National Supercomputing Center (CSCS) under project s575.

## A   Appendix

For reference, the Fermi surfaces of the nondisproportionated, nonmagnetic $R\bar{3}c$ and $Pnma$ phases are give in Figs. 10 and 11, respectively.

The relaxed atomic positions of the disproportionated antiferromagnetic $R$ and $P$ phases are given in Tables 2 and 3, respectively. In the tables, Ni$_↑$, Ni$_↓$, and Ni$_0$ denote Ni sites with spins up, down, and zero, respectively.

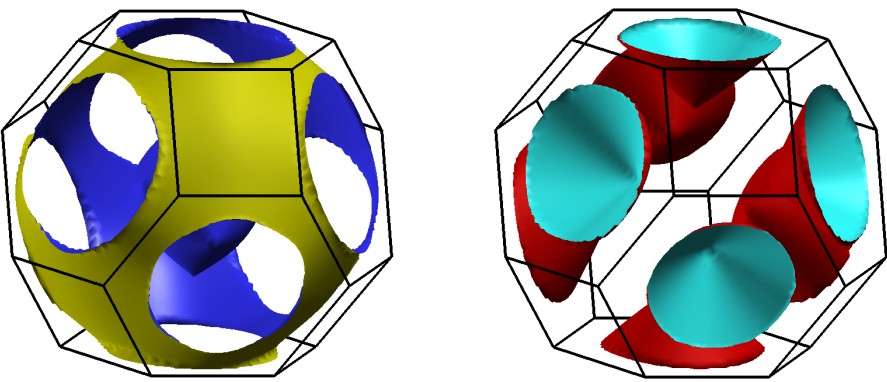

Figure 10: Calculated Fermi sheets of LaNiO$_3$ in the nondisproportionated, nonmagnetic $R\bar{3}c$ phase.

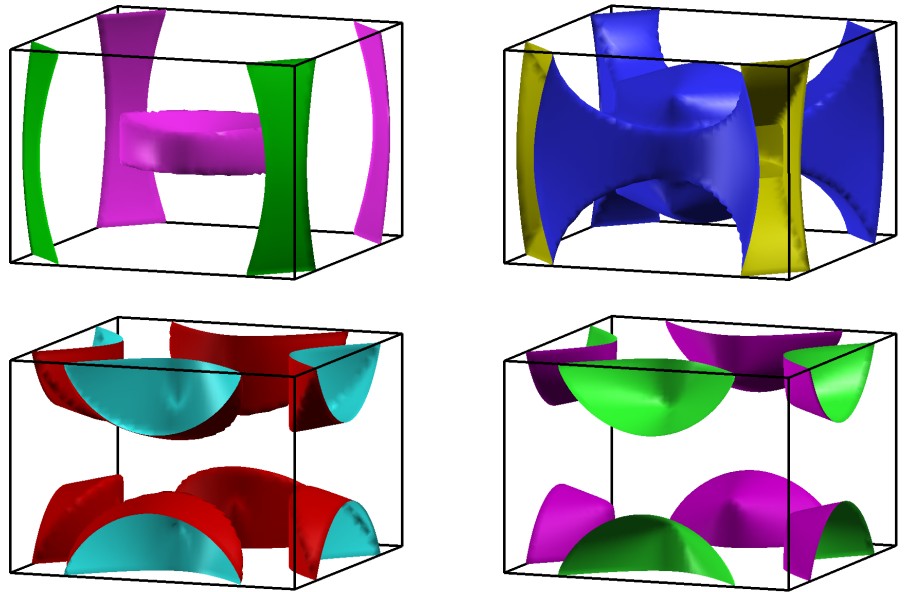

Figure 11: Calculated Fermi sheets of LaNiO$_3$ in the nondisproportionated, nonmagnetic $Pnma$ phase.

Table 2: The relaxed atomic positions of the disproportionated antiferromagnetic $R$ phase. The atomic positions are given relative to the primitive lattice vectors. The lattice parameters are $a = 5.4064, b = 5.4078, c = 10.8092$ Å, $\alpha = 61.306°, \beta = 61.315°$, and $\gamma = 61.167°$.

| atom | $x$ | $y$ | $z$ |
|---|---|---|---|
| La | 0.7502 | 0.7494 | 0.3751 |
| La | 0.7502 | 0.7494 | 0.8751 |
| La | 0.2498 | 0.2506 | 0.1249 |
| La | 0.2498 | 0.2506 | 0.6249 |
| $Ni_{\uparrow}$ | 0.0000 | 0.0000 | 0.0000 |
| $Ni_{\downarrow}$ | 0.0000 | 0.0000 | 0.5000 |
| $Ni_0$ | 0.5000 | 0.5000 | 0.2500 |
| $Ni_0$ | 0.5000 | 0.5000 | 0.7500 |
| O | 0.2486 | 0.8070 | 0.3455 |
| O | 0.2486 | 0.8071 | 0.8454 |
| O | 0.7514 | 0.1929 | 0.1545 |
| O | 0.7514 | 0.1930 | 0.6545 |
| O | 0.8042 | 0.6936 | 0.1253 |
| O | 0.8042 | 0.6936 | 0.6253 |
| O | 0.1958 | 0.3064 | 0.3747 |
| O | 0.1958 | 0.3064 | 0.8747 |
| O | 0.6912 | 0.2553 | 0.4033 |
| O | 0.6912 | 0.2553 | 0.9033 |
| O | 0.3088 | 0.7447 | 0.0967 |
| O | 0.3088 | 0.7447 | 0.5967 |

Table 3: The relaxed atomic positions of the disproportionated antiferromagnetic $P$ phase. The atomic positions are given relative to the primitive lattice vectors. The lattice parameters are $a = 10.8835, b = 15.4253, c = 5.4979$ Å, $\alpha = 90.028°, \beta = 89.977°$, and $\gamma = 90.070°$.

| atom | $x$ | $y$ | $z$ | atom | $x$ | $y$ | $z$ |
|---|---|---|---|---|---|---|---|
| La | 0.4880 | 0.3751 | 0.9955 | O | 0.2474 | 0.1244 | 0.9367 |
| La | 0.4880 | 0.8751 | 0.9955 | O | 0.2474 | 0.6244 | 0.9367 |
| La | 0.9880 | 0.3751 | 0.9955 | O | 0.7474 | 0.1244 | 0.9367 |
| La | 0.9880 | 0.8751 | 0.9955 | O | 0.7474 | 0.6244 | 0.9367 |
| La | 0.2381 | 0.3750 | 0.5044 | O | 0.4975 | 0.1256 | 0.5634 |
| La | 0.2381 | 0.8750 | 0.5044 | O | 0.4975 | 0.6256 | 0.5634 |
| La | 0.7381 | 0.3750 | 0.5044 | O | 0.9975 | 0.1256 | 0.5634 |
| La | 0.7381 | 0.8750 | 0.5044 | O | 0.9975 | 0.6256 | 0.5634 |
| La | 0.0120 | 0.1249 | 0.0045 | O | 0.3863 | 0.0171 | 0.2258 |
| La | 0.0120 | 0.6249 | 0.0045 | O | 0.3863 | 0.5171 | 0.2258 |
| La | 0.5120 | 0.1249 | 0.0045 | O | 0.8863 | 0.0171 | 0.2258 |
| La | 0.5120 | 0.6249 | 0.0045 | O | 0.8863 | 0.5171 | 0.2258 |
| La | 0.2620 | 0.1250 | 0.4956 | O | 0.1137 | 0.4829 | 0.7743 |
| La | 0.2620 | 0.6250 | 0.4956 | O | 0.1137 | 0.9829 | 0.7743 |
| La | 0.7620 | 0.1250 | 0.4956 | O | 0.6137 | 0.4829 | 0.7743 |
| La | 0.7620 | 0.6250 | 0.4956 | O | 0.6137 | 0.9829 | 0.7743 |
| Ni$_\uparrow$ | 0.0000 | 0.0000 | 0.5000 | O | 0.3880 | 0.2329 | 0.2268 |
| Ni$_\uparrow$ | 0.5000 | 0.5000 | 0.5000 | O | 0.3880 | 0.7329 | 0.2268 |
| Ni$_\uparrow$ | 0.2500 | 0.2500 | 0.0000 | O | 0.8880 | 0.2329 | 0.2268 |
| Ni$_\uparrow$ | 0.7500 | 0.7500 | 0.0000 | O | 0.8880 | 0.7329 | 0.2268 |
| Ni$_\downarrow$ | 0.0000 | 0.4500 | 0.5000 | O | 0.1120 | 0.2671 | 0.7732 |
| Ni$_\downarrow$ | 0.5000 | 0.0000 | 0.5000 | O | 0.1120 | 0.7671 | 0.7732 |
| Ni$_\downarrow$ | 0.2500 | 0.7500 | 0.0000 | O | 0.6120 | 0.2671 | 0.7732 |
| Ni$_\downarrow$ | 0.7500 | 0.2500 | 0.0000 | O | 0.6120 | 0.7671 | 0.7732 |
| Ni$_0$ | 0.0000 | 0.2500 | 0.5000 | O | 0.3628 | 0.4827 | 0.7284 |
| Ni$_0$ | 0.5000 | 0.7500 | 0.5000 | O | 0.3628 | 0.9827 | 0.7284 |
| Ni$_0$ | 0.7500 | 0.0000 | 0.0000 | O | 0.8628 | 0.4827 | 0.7284 |
| Ni$_0$ | 0.2500 | 0.5000 | 0.0000 | O | 0.8628 | 0.9827 | 0.7284 |
| Ni$_0$ | 0.2500 | 0.0000 | 0.0000 | O | 0.1372 | 0.0173 | 0.2716 |
| Ni$_0$ | 0.7500 | 0.5000 | 0.0000 | O | 0.1372 | 0.5173 | 0.2716 |
| Ni$_0$ | 0.0000 | 0.7500 | 0.5000 | O | 0.6372 | 0.0173 | 0.2716 |
| Ni$_0$ | 0.5000 | 0.2500 | 0.5000 | O | 0.6372 | 0.5173 | 0.2716 |
| O | 0.2526 | 0.3756 | 0.0633 | O | 0.3629 | 0.2671 | 0.7240 |
| O | 0.2526 | 0.8756 | 0.0633 | O | 0.3629 | 0.7671 | 0.7240 |
| O | 0.7526 | 0.3756 | 0.0633 | O | 0.8629 | 0.2671 | 0.7240 |
| O | 0.7526 | 0.8756 | 0.0633 | O | 0.8629 | 0.7671 | 0.7240 |
| O | 0.0025 | 0.3744 | 0.4366 | O | 0.1371 | 0.2329 | 0.2760 |
| O | 0.0025 | 0.8744 | 0.4367 | O | 0.1371 | 0.7329 | 0.2760 |
| O | 0.5025 | 0.3744 | 0.4366 | O | 0.6371 | 0.2329 | 0.2760 |
| O | 0.5025 | 0.8744 | 0.4366 | O | 0.6371 | 0.7329 | 0.2760 |

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
