# Peer review of "Breathing distortions in the metallic, antiferromagnetic phase of LaNiO$_3$"

_SciPost Physics, doi:SciPost Phys. 5, 020 (2018)_

## Round 1 · Referee Report · Markus Aichhorn · 2017-10-17

Strengths

1) This paper is a very thorough study of the structural and magnetic properties of the rare-earth nickelate LaNiO3, including a comprehensive comparison to other members of the nickelate family such as YNiO3. Alaska Subedi did take a lot of care in analysing the structural instabilities of the system, and gave an understandable explanation of how LaNiO3 is different to the other compounds.
2) The paper is well placed in the context of other current works, and discusses all relevant developments appropriately.
3) Alaska Subedi also openly discusses the failures of his approach, in the sense that he does not only tell what works. For instance, it is valueable that he mentions also structures that he tries to stabilize and did not succeed to do so.

Weaknesses

1) Given that Alaska Subedi did do many calculations, it is in some parts not so easy to follow the arguments. For example, I had to read page 8, which discusses all the different possible magnetic orderings and corresponding unit cells, several times until I got the point. At the current stage, I would encourage Alaska Subedi to make in this part more clear what he is aiming at, before entering all the details of the results. Reading these paragraphs would be much more easy if one knows what the main message should be.

Report

I think that this is a very timely work that discusses a very interesting compound, based on state-of-the-art calculations. It tries to convey a complete picture of how structure and magnetism is intertwined, and how this is reflected in the electronic properties.

I have a concern about the accuracy of the calculations. At several points, Alaska Subedi discusses tiny differences of, e.g., total energies, for instance the 1eV difference of the Pnma and R3c structure of LaNiO3 (Table 1). As a matter of fact, these numbers have no real, or very difficult to estimate, error bars. In principle, they should be converged in several parameters, most importantly the energy cut-off and the k-space integration. I understand that this is complicated, and computational costs are prohibitive. Nevertheless, a discussion of the accuracy of the given numbers is necessary.

When comparing to the experimental works, two things come to my mind: PBE gives the wrong structure as the ground state, and stabilizes a magnetic ordering that is at least strongly debated. The first issue could be related to the above given possible problems with convergence. The second, however, needs a bit more discussion. As far as I understood the argument, the rock-salt-type breathing distortions come hand in hand with charge-disproportionation and the up-0-down-0 magnetic ordering. Now the question: Is it possible to stabilize this structure with the distortions also without magnetic ordering? This would strengthen the argument that magnetic ordering is not detrimental for the phase transition, but rather follows the effects in the structure.

Two general remarks on section 3.3: First, matrix elements are neglected in the Lindhard formula. We did some work recently showing that matrix elements can have a significant effect on the relative height of excitations in the Brillouin zone (Heil et al., Phys.Rev.B 90, 115142, 2014). Maybe some comments on this could be added here. Second, magnetic instabilities should follow from the RPA-enhanced suszeptibility, and not the bare one. This is also discussed in above paper, where also references to work by the group of Hardy Gross can be found, which studied this problem in the context of iron-chalcogenide superconductors.
In general, I am not sure if this section 3.3 is absolutely necessary for the manuscript.

Requested changes

1) Some discussion on the numerical accuracy of the total energies.
2) Some rewriting, in particular in section 3.2, to make the main points clearer.
3) It would be nice if Alaska could clarify if a disproportionated state exists without magnetic ordering.

---

## Round 1 · Referee Report · Anonymous · 2017-12-1

Strengths

1 - very thorough study

2 - useful for other researchers as it gives many details

3 - deals with Ni-oxide compounds, which are currently of big interest

Weaknesses

1 - the encyclopedic tone makes it a big hard to read

Report

This manuscript focuses mainly on LaNiO3 and YNiO3 which are analyzed within DFT. It studies the magnetic phases, calculates the Fermi surfaces and band structures and makes contact with a number of recent experiments on LaNiO3.

I have read the comments by the other referee and I agree essentially with all of them. I would like to add only the following point:

When discussing DFT+DMFT results, it would be good to mention that they depend crucially on the filling of the e_g manifold. The author cites Park et al. [19] and a study by himself [20]. In another study [PHYSICAL REVIEW B 88, 195116 (2013)] a model calculation showed that the relative energy position of the x2-y2 and z2 orbitals strongly depends on whether the low-energy model is close to quarter or half filling. This can be explained invoking a weaker or stronger effect of the Hund's coupling, respectively. The number of electrons sitting in the e_g manifold depends on how strong the hybridization to the p-ligand orbitals is. In the absence of p-bands (e_g only model) the model is quarter-filled and J_Hund influences the physics very little. This results in a large energy separation between x2-y2 and z2. The more electrons are donated to the e_g from the p-orbitals within a dp-model, the stronger J_Hund can operate and x2-y2 and z2 as a consequence are less separated and both contribute to the Fermi surface.
This aspect should be discussed to some extent, in my opinion.

Requested changes

1 - add the comment on the DFT+DMFT calculation

---

## Round 2 · Author Response

I thank the anonymous referee and Prof. Aichhorn for the review and their positive evaluation. I also thank them for suggestions for changes. I have addressed their issues, which I detail below.

Anonymous referee: 1 - add the comment on the DFT+DMFT calculation. I have added a reference to the mentioned DFT+DMFT calculations.

Prof. Aichhorn: 1) Some discussion on the numerical accuracy of the total energies. A new paragraph is added discussing the numerical accuracy.

2) Some rewriting, in particular in section 3.2, to make the main points clearer. Section 3.2 has been reorganized. A paragraph has been deleted and subsections have been added.

3) It would be nice if Alaska could clarify if a disproportionated state exists without magnetic ordering. I have clarified this.

---

## Round 2 · List of Changes

List of changes:

1) The arXiv link to the original experimental paper has been updated with the reference to the published paper.

2) Ref. 41, Parragh et al. [PRB 88, 195116 (2013)] has been added in page 4.

3) I have added a following paragraph at the end of Methods section:

"To check convergence with respect to the planewave cut-off, I repeated the structural relaxations of the various Glazer tilts for a cut-off value of 60 Ry. Same energetic rankings were obtained. I also did some calculations with larger $k$-point meshes, and this did not change the results in a meaningful way. Note that the meshes used in this work are denser than those used in two recent DFT studies on the rare-earth niclelates \cite{Varignon2017,Hampel2017}."

4) The imaginary i have been added for the imaginary frequencies in page 6.

5) The a+b-c- tilt is not stabilized for YNiO3, and the corresponding entry in Table I is changed to "---".

6) Section 3.2 is split into two sections: 3.2.1 Spin-polarized structural relaxations in $R\overline{3}c$ LaNiO$_3$ 3.2.2 Spin-polarized structural relaxations in $Pnma$ LaNiO$_3$

7) The paragraph "An intriguing aspect of..." near the beginning of section 3.2 has been completely removed to increase clarity.

8) Motivation for the multitude of the spin-polarized calculations have been provided by adding the following paragraph in page 8:

"To understand the nature of magnetic instabilities, if there are any, in LaNiO$_3$ and possible competition between different magnetic interactions, I extensively studied the stability of diverse magnetic ordering phases in several supercells of $R\overline{3}c$ and $Pnma$ structures using spin-polarized DFT calculations within the PBE GGA."

9) The negligence of the matrix element in the Lindhard susceptibility is discussed and justified in futher detail in page 10:

"The negligence of the matrix element changes the relative intensities of the peaks in the susceptibility \cite{Heil2014}. However, major features remain the same and qualitative understanding can still be gleaned off from such an approximation. I note that a previous discussion of the magnetic susceptibility in the nickelates has also made this approximation \cite{Lee2011b}."

11) I have added the following sentence at the end of section 3.2:

"This is also supported by the fact that I was not able to stabilize breathing distortions in the non-spin-polarized calculations."

10) In Fig. 7, the "L-AFM" and "T-AFM" have been changed to "R-AFM" and "P-AFM", respectively.

11) The last two paragraphs in page 13 have been slightly reworked to improve readability.

---

## Editorial Decision

published